# Predicting Three-Dimensional Dose Distribution of Prostate Volumetric Modulated Arc Therapy Using Deep Learning

**DOI:** 10.3390/life11121305

**Published:** 2021-11-27

**Authors:** Patiparn Kummanee, Wares Chancharoen, Kanut Tangtisanon, Todsaporn Fuangrod

**Affiliations:** 1Princess Srisavangavadhana College of Medicine, Chulabhorn Royal Academy, Bangkok 10210, Thailand; 62020208@pccms.ac.th (P.K.); wares.cha@cra.ac.th (W.C.); 2Department of Computer Engineering, King Mongkut’s Institute of Technology Ladkrabang, Bangkok 10520, Thailand; ktkanut@kmitl.ac.th

**Keywords:** VMAT, deep learning, prostate cancer, dose distribution prediction, generative adversarial network

## Abstract

Background: Volumetric modulated arc therapy (VMAT) planning is a time-consuming process of radiation therapy. With a deep learning approach, 3D dose distribution can be predicted without the need for an actual dose calculation. This approach can accelerate the process by guiding and confirming the achievable dose distribution in order to reduce the replanning iterations while maintaining the plan quality. Methods: In this study, three dose distribution predictive models of VMAT for prostate cancer were developed, evaluated, and compared. Each model was designed with a different input data structure to train and test the model: (1) patient CT alone (PCT alone), (2) patient CT and generalized organ structure (PCTGOS), and (3) patient CT and specific organ structure (PCTSOS). The generative adversarial network (GAN) model was used as a core learning algorithm. The models were trained slice-by-slice using 46 VMAT plans for prostate cancer, and then used to predict and evaluate the dose distribution from 8 independent plans. Results: VMAT dose distribution was generated with a mean prediction time of approximately 3.5 s per patient, whereas the PCTSOS model was excluded due to a mean prediction time of approximately 17.5 s per patient. The highest average 3D gamma passing rate was 80.51 ± 5.94, while the lowest overall percentage difference of dose-volume histogram (DVH) parameters was 6.01 ± 5.44% for the prescription dose from the PCTGOS model. However, the PCTSOS model was the most reliable for the evaluation of multiple parameters. Conclusions: This dose prediction model could accelerate the iterative optimization process for the planning of VMAT treatment by guiding the planner with the desired dose distribution.

## 1. Introduction

Prostate cancer was the second most diagnosed cancer and the fifth leading cause of death among the male population globally in 2020 [1]. External-beam radiation therapy (EBRT) is one of the most widely used treatment modalities for prostate cancer. Volumetric modulated arc therapy (VMAT) and intensity-modulated radiation therapy (IMRT) are also widely used, and have become the standard for prostate cancer treatments in many institutes [2,3]. In VMAT and IMRT, the complex dose distribution can increase the dose conformity to the target and significantly decrease the amount of dose given to the organs at risk (OARs), which can reduce the risk of complications to normal tissues after treatment [4]; however, in order to achieve a higher complex dose distribution, the complexity of the treatment planning process is also increased [4,5]. VMAT treatment planning is considered a time-consuming process due to the manual nature of the process (trial-and-error inverse planning), the long dose calculation time, and the replanning iterations. In order to achieve the desired dose distribution, the dosimetrist or the planner has to manually input the optimized parameters to the treatment planning system (TPS). After that, the system will calculate and evaluate the dose distribution outcome through dose prescription and dose criteria via planning target volume (PTV) and OARs, respectively. If the evaluation outcomes do not satisfy the criteria, the process of adjusting these planning parameters is repeated, and the process of repeating repetition may take time for each cycle. The process may be iterated from an unapproved plan, and more requirements may be needed from the radiation oncologist’s perspective.

In recent years, researchers have started to use deep learning and neural networks (NNs) in various medical and biomedical applications [6]. In biomedical tasks, deep learning has been used as an identification and detection algorithm. For instance, deep learning has been applied in the detection of protein S-sulfenylation sites from protein sequences [7]. Another pertinent point is that protein function identification provides a better understanding of cancer and helps to create more effective drugs for cancer treatment [8]. Additionally, deep learning can be used for protein–protein interaction, which uses text mining methods from biomedical literature [9]. Deep learning is also employed to accelerate and assist the treatment planning process in various applications, one of which is predicting dose distribution. Deep learning can predict dose distributions by inputting anatomical information (such as CT images or contours of organ structures) as either 2D or 3D data. The predicted dose distribution can be used as an objective to automatically generate a treatment plan later in the process [10]. Nguyen et al. studied the use of the U-net—a deep learning model—in dose distribution prediction of IMRT prostate cancer plans; they reported that when comparing the model and ground truth, the results showed an average absolute difference within 5% of the prescription dose, and had Dice similarity of 0.91 [5]. Murakami et al. developed a fully automated dose distribution prediction of IMRT for prostate cancer using a GAN, which requires only CT images to generate the prediction dose. The results show that the dose differences evaluated by the DVH parameters are approximately 2% of the prescription dose for PTV and 3% for OARs, except for the parameters D_98%_ and D_95%_ [4]. Fan et al. developed an automated treatment planning strategy using a residual neural network deep learning model (ResNet). In particular, ResNet can predict the 3D dose distributions and use MATLAB to generate a treatment plan based on the predicted dose distributions. The authors reported that the model could predict the acceptable dose distributions with no statistical differences between the actual and predicted plans [11]. Mahmood et al. proposed the GAN model to predict the 3D dose distribution of oropharyngeal cancer, and compared the predicted results with several baseline approaches. They reported that the plan generated by the GAN-predicted dose distribution outperformed the actual plans by satisfying the additional OAR criteria, achieving better results than other baseline methods [12].

However, the dose distribution prediction system for VMAT plans based on deep learning has not been widely investigated. In order to solve the time-consuming problem of the VMAT treatment planning process, this study aims to develop a method of predicting 3D dose distribution for prostate VMAT based on deep learning. This would accelerate the treatment planning process by guiding the planner with the desired dose distribution without a dose calculation from TPS. This study used a deep learning algorithm to investigate the accuracy of learning techniques with customized input sources: (1) patient CT alone (PCT alone), (2) patient CT and generalized organ structure (PCTGOS), and (3) patient CT and specific organ structure (PCTSOS). The core learning algorithm uses the generative adversarial network model. The dose distribution was predicted using 2D images, which were compiled by stacking each 2D slice to create the 3D dose distribution. The baseline of this study was the dose distribution that was calculated from the actual treatment plan. The prediction accuracy of the customized input sources model was evaluated with the ground truth dose distribution using 3D gamma analysis and the difference in DVH parameters. Gamma analysis is a technique of evaluating the accuracy of the dose distributions by simultaneously considering the dose difference (DD) and the distance to agreement (DTA) [13]. The distribution comparison was evaluated using the acceptance criteria. The dose and distance of the two compared distributions were normalized by dose difference criteria (%) and distance criteria (mm), based on the clinical standard. If the evaluated point on the gamma index is higher than 1, the point does not satisfy the criteria. The DVH parameters of PTV and OARs are the clinical plan evaluation parameters; in this case, the physician and medical physicist used these to evaluate the plan before the treatment process. The parameters of PTV are the clinical criteria of the prescription dose for each specific target organ. The parameters of OARs are the dose constraint recommendations from Quantitative Analysis of Normal Tissue Effects in the Clinic (QUANTEC) and Radiation Therapy Oncology Group (RTOG) clinical trials, consisting of the bladder, the rectum, and both the left and right sides of the femoral head [14,15]. The dose constraints recommended by QUANTEC for the bladder and rectum are shown in Table 1 [16,17]. For the left and right femoral head, dose constraints are recommended by RTOG, as shown in Table 1 [15].

## 2. Materials and Methods

The overall framework of this study is shown in Figure 1. The framework is separated into three main processes: a model construction, a dose distribution prediction, and a model evaluation and comparison. The model construction process is the process of constructing three prostate VMAT dose distribution prediction models involving different input structures using the GAN deep learning model, wherein, each model is trained with a specific input data structure from a training dataset. The following process constitutes the dose distribution prediction: Dose distributions were predicted from each trained model using a specific testing data structure. The model evaluation and comparison are the processes of evaluating each trained model’s prediction accuracy with the ground truth dose distribution using 3D gamma analysis. The epoch that gives the maximum gamma passing rate of each model was chosen, so as to be the representative model to be evaluated using various DVH parameters. Then, accuracy was compared between the three models.

### 2.1. Data Acquisition

Data from 54 prostate cancer patients treated at Chulabhorn Oncology Medical Center, Chulabhorn Hospital, Thailand, between 2015 and 2020, were used in this study. The data consisted of patient CT images, the contours of structures, and dose distribution. All patients were treated using the VMAT technique with a prescription dose of 78 Gy/39 fractions to a PTV. A total of 48 patients’ data (85.71%) were used as the training dataset, with a total of 5980 image slices, and the remaining 8 patients (14.29%) were used as the test dataset, with a total of 1098 image slices.

The PTV is separated into three volumes that receive a different prescription dose. The first volume receives 46 Gy (PTV46) or 23 fractions, which irradiate to the prostate, the seminal vesicle, and the lymph nodes around the pelvic area. The next volume is 14 Gy or 7 fractions, which irradiate the whole prostate and seminal vesicle. In particular, this means that the whole prostate and seminal vesicle would receive a total of 60 Gy (PTV60). The following 18 Gy or 9 fractions would irradiate only the prostate area. Thus, the prostate would receive a total 78 Gy (PTV78) from all treatment processes.

### 2.2. Data Pre-Processing

The acquired dose distributions (dose distribution of PTV46, -60, and -78) were combined to receive one input dose distribution pairing with one corresponding CT image and contour of organ structure. For the organ structure of the PTV, the largest PTV (PTV46) that covers all treatment targets was chosen. Each slice of CT image and organ structure was resized from 512 × 512 pixels to 256 × 256 pixels in order to reduce memory usage. Dose distribution with the patient’s specific size was cropped and resized to be the same as CT images (256 × 256 pixels). All CT images, organ structures, and dose distribution slices were normalized from –1 to 1.

### 2.3. Dataset Generation

Each model was designed with a different purpose, and required different input data structures to train and test the model. The PCT alone model was designed to generate the prediction model from only CT images, without using any contour structure data. The resized CT images and resized dose distribution were paired with each corresponding CT and its dose distribution to create the dataset, as shown in Figure 2a. PCTGOS requires the contours of organ structure and CT images as the input data, along with dose distribution as the target images. The organ structures consist of 5 volumes, including PTV, body, and 3 organs at risk (bladder, femur, and rectum), as shown in Figure 2b. In addition, PCTSOS consists of 5 models with the same architecture, but different input data. The dataset includes 5 input sources consisting of the organ structure and CT images as the input data, along with dose distribution as the target images, similar to the previous model. Nevertheless, the organ structure and dose distribution were split into 5 volumes, including the PTV, body, and 3 OARs (bladder, femur, and rectum), as shown in Figure 2c. After that, all CT images, organ structures, and dose distribution were normalized from −1 to 1, and each corresponding CT, organ structure, and its dose distributions were paired to create organ-specific datasets.

### 2.4. Generative Adversarial Model and Dose Distribution Prediction

In this study, the modified GAN model called pix2pix was used to construct the predictive model. The pix2pix model is a specific GAN model for image-to-image translation proposed by Isola et al. [18]. The pix2pix model can generate images by learning the dataset of the pairs of the two images. The CT images and contour structures are considered the input or source images. The corresponding dose distributions are the labels or target images. Goodfellow et al. proposed a GAN model that was constructed using two neural network models: the generator, and the discriminator [19]. In this case, the generator was trained to generate the dose distributions that could not be discriminated against by the discriminator; the discriminator was trained to maximize the probability of discrimination between the generated dose distribution from the generator and the ground truth. A U-net-based architecture model was used as the generator, consisting of 8 downsampling convolutions and 8 upsampling convolutions with an input size of 256 × 256 pixels and the same output size. The PatchGAN classifier was used as the discriminator to determine whether each image patch was real or not. The model consisted of 5 downsampling convolutions with an output size of 30 × 30 patches.

The prediction models were trained using an HP Z8 G4 workstation with an Intel Xeon 4112 and a NVIDIA Quadro P4000 GPU. Both the generator and the discriminator used the binary cross-entropy as the cost function and the rectified linear unit (ReLU) as the activation function. We used an Adam optimizer as the optimizer [20], with a learning rate of 0.0001, and the momentum parameters were β1 = 0.5 and β2 = 0.999. We used the default Adam momentum parameters described by Isola et al. [18], as they were proven to be great for image-to-image translation problems. The training batch size was set to 4, based on the RAM of the GPU. Based on the preliminary experiments, the models were trained with 100, 200, 300, 400, and 500 epochs.

The dose distribution results were predicted by inputting a testing dataset created specifically for each model. In order to generate the dose distribution from the PCTSOS model, each organ-specific testing dataset was inputted to each model. After generating all organ-specific dose distributions, the full-body dose distribution was derived by summing all predicted dose distributions.

### 2.5. Model Evaluation

The accuracy of the predicted dose distribution was assessed using 3D gamma analysis for each epoch of each model with the 3%/3 mm criteria. The epoch that gave the maximum gamma passing rate of each model was selected to obtain the representative results to compare the prediction performance between the three models.

The dose differences were calculated for the evaluation parameters for PTV (Table 2) and OARs (Table 3), which are the dose constraint recommendations from the QUANTEC and RTOG clinical trials.

The differences in the DVH evaluation parameters were calculated using the following equation, based on the previous studies of Murakami et al. and Nguyen et al. [4,5]:(1)$$\mathrm{Absolute}\;\mathrm{dose}\;\mathrm{difference}=\frac{{|\;D}_{\mathrm{prediction}}-{\;D}_{\mathrm{ground}\;\mathrm{truth}}\;|}{D_{\mathrm{prescription}}}\;\times \;100,$$
where D_prediction_ represents the DVH parameters from the predicted dose distribution, D_ground truth_ represents the DVH parameters from the actual dose distribution, and D_prescription_ represents the prescription dose to the PTV. In this case, the difference in the DVH parameters, which are measured in terms of volume, was calculated using the following equation:Absolute volume difference = | V_prediction_ − V_ground truth_ |,(2)
where V_prediction_ is the percentage of the organ’s volume from the predicted dose distribution, and V_ground truth_ is the percentage of the organ’s volume from the ground truth dose distribution.

The uncertainty of this study was evaluated by the standard deviation (±SD) of the results for each model. For instance, the prediction time of the PCT alone model was 24.87 s/8 data, with the uncertainty reported as (±SD) with 3.61 ± 0.19 s/data.

## 3. Results

### 3.1. Training and Prediction Time

The training and prediction times are shown in Table 4. For the PCT alone model and the PCTGOS model, the mean prediction time was approximately 3.5 s per patient. In contrast, the PCTSOS model spent approximately 17.5 s per patient.

### 3.2. 3D Gamma Analysis

Table 5 shows the average 3D gamma passing rate with 3%/3mm criteria. The maximum average gamma passing rate was 80.51 ± 5.94% from patient CT, including a generalized organ structure model with a training stage of 400 epochs, whereas the maximum gamma passing rate of the patient CT alone model was 77.21 ± 9.02% from 200 epochs, and that of the patient CT and specific organ structure model was 76.90 ± 3.91% with 300 epochs.

The result after the gamma analysis part represents each model via the epoch with maximum gamma passing rate. Models include the PCT alone model at 200 epochs, the PCTGOS model at 400 epochs, and the PCTSOS model at 300 epochs.

The example results of three dose distribution prediction models of patient number five are shown in Figure 3. Dose profile comparisons between ground truth dose distribution and predicted dose distribution for all three models are shown in Figure 4. The example results were from slice number 60 (middle slice). Dose profiles in all three models were highly consistent with the ground truth. However, in the PCTSOS model, the dose profile lacked smoothness due to an inconsistency of the prediction model in specific organs.

### 3.3. DVH Parameters Evaluation

Example results of comparisons of DVH parameters between ground truth dose distribution and predicted dose distribution are shown in Figure 5. The example results were from patient number 5, slice number 60 (middle slice).

The summary of percentage differences between the average DVH parameters of ground truth dose distribution and predicted dose distribution for all three models is shown in Table 6. In PTV78 and PTV60, the PCTSOS model showed the best results in terms of the average percentage dose difference, when compared with the other two models (except for the parameter D2% of PTV78 from the CT alone model). However, the model PCTGOS in PTV46 appeared to have the lowest percentage difference, when all three models were compared. In this case, in PTV46, the patient CT alone model had the highest percentage difference, with approximately 20%. The PCTGOS model and the PCTSOS model were lower in terms of percentage difference in ground truth, at approximately 3.5% and 10%, respectively.

The PCTSOS model showed the highest performance for the bladder, with the lowest percentage difference (approximately 4.5%) between the ground truth and predicted dose distributions. On the other hand, the PCTGOS model had the best agreement in the rectum, with approximately 10% dose difference. In addition, when comparing all three models, the PCTGOS model showed the most promising percentage of dose difference in both the left and right femoral heads. The percentage volume differences of both the left and right femoral heads contained similar values.

### 3.4. Model Comparison

The summary of the model comparison is shown in Table 7. The prediction time of each model was approximately 3.5 s. However, for the PCTSOS model, which comprises five organ models, the cumulative prediction time was approximately 17 s. The patient CT alone model had a 3D gamma passing rate of 77.21 ± 9.02 with the 3/3mm criteria—only slightly lower than the best result from the PCTGOS model (80.51 ± 5.94). The average overall percentage difference in the parameters of the PCTGOS model was 6.01 ± 5.44%, which is slightly lower those of than the other two models. The acceptance of the model comparison criteria counted the number of lowest percentage dose differences between the ground truth and predicted dose distribution. The PCTSOS model has the best prediction model accuracy, as it contains the best 11 parameters, including 6 from PTV and 5 from OARs. The PCT alone model showed the lowest prediction model accuracy, including 1 from PTV and 4 from OARs. In addition, the PCTGOS model contains 10 parameters, including 2 from PTV and 8 from OARs.

## 4. Discussion

In this study, several dose distribution predictive models of VMAT for prostate cancer were developed, evaluated, and compared with the core concept of backward planning. The traditional treatment process starts from patient simulation, organ structure contouring, and treatment planning stages to obtain the dose distribution and evaluate the planning. In contrast, our method can skip the planning and evaluation stages to achieve the desired dose distribution without dose calculation via the system. This can reduce the planning iterations and planning time by guiding the planner with the acceptable dose distribution, and can be the objective of the automated treatment planning based on the predicted dose distribution. In addition, the treatment planning time is dramatically reduced for the PCT alone model, which can skip the organ-structure-contouring process.

In this study, the planning process included two full arcs of VMAT and three PTVs (PTV78, PTV60, PTV46) that were used in all patients to deliver radiation to the prostate. This gave uniformity and a great pattern to the dataset, making it suitable for training the model. However, applying the model to predict the dose distribution of patients who are treated with other techniques may not give the best results—even in prostate cancer patients.

The data used in this study were obtained from a single hospital center, which may lead to a bias in this hospital data. Applying this model to different hospital centers may not give the best performance based on the dependence variation of the model. However, the dataset that was used in this model was analyzed by several experienced physicians and medical physicists who used the same criteria for the treatment planning process. This should prevent bias of the data from the treatment planner, and may also increase the generalizability of the model. Note that the number of training datasets could influence the prediction accuracy in deep learning algorithms. In the future, our study could be improved by increasing the number of training datasets.

Although the prediction results of the PCTSOS model had the lowest percentage differences in the DVH parameters (11 parameters), the 3D gamma analysis and average percentage difference over all parameters still showed worse results than those of the PCTGOS model. Additionally, the PCTSOS model required more prediction time than the PCTGOS model for the prediction of five organ-specific models. Compared to the PCT alone model, the prediction time of the PCTGOS model is comparable, but the PCT alone model does not require the contours of organ structure, meaning that the PCT alone models required less time than the PCTGOS model. The PCTGOS model gave the best predictive results of the three models, while the PCT alone model gave the worst. However, the PCT alone model seems to be interesting for further investigation in order to develop fully automated dose prediction without using the patient organ structure information.

In the PCTSOS model, dose distribution lacked smoothness around the edge of the organs. This may be because the dose distribution of each organ comes from a different predictive model that trains from the specific organ data. The dose distribution of each organ was predicted separately and then combined with other organs into a full-body dose distribution, which caused a discontinuous dose between the organs’ boundaries. To solve this problem, we suggest applying filters to a full-body dose distribution, such as a Gaussian blur, in order to reduce the discontinuity between each organ dose.

The patient data used in this study consisted of three PTVs: one larger, and two smaller inside. We only chose the largest PTV to be the representative volume of the high-dose volume. The results show that, in PTV46, the patient CT alone model shows the highest percentage differences (approximately 20%), compared to the model where organ structures were included. The patient CT and specific organ structure model were lower, at approximately 3.5% and 10% differences, respectively. This means that the area of steep dose gradient can strongly influence the organ structures. Thus, in future work, all PTVs in the sequential plan should be implemented in order to improve the accuracy of the predictive model.

The discriminator used in this study was a 30 × 30 PatchGAN. In the previous study of Isola et al. [18], the authors compared four sizes of PatchGAN, consisting of 1 × 1, 16 × 16, 70 × 70, and 286 × 286 patches, and reported that the 70 × 70 patches gave the most accurate results. The various sizes of PatchGAN should be included in future studies. The 3D dose distribution was processed from a 2D slice-by-slice dose prediction method in which the information from the vertical axis of the patient’s body was not implemented. This predicted dose distribution lacks smoothness in the vertical axis, and reduces the accuracy at the upper and lower borders of the PTV and OARs. In contrast, in the study of Dan Nguyen et al. [21], which studied the 3D dose distribution prediction from a hierarchically densely connected U-net, the authors reported that the predicted dose distribution from the 3D HD U-net gave better performance compared to a standard 2D U-net in terms of D_max_, homogeneity, dose conformity, and dose coverage.

Willems et al. studied the use of a 3D U-Net-based deep learning model to predict the dose distribution for VMAT for prostate cancer from CT images alone. Accordingly, they compared their results to CT images that were combined with additional data (plan isocenter and contours of organ structures). They reported on the CT only model that the mean percentage error in D_max_ and D_98%_ was 8.6% and 16.8%, respectively, whereas the CT combined with the contour structure model resulted in a decrease in mean percentage error in D_max_ and D_98%_ to 1.3% and 1.0%, respectively [22]. Comparing these previous studies to our results, as shown in Table 8, the absolute difference in D_max_ from our PCT alone model was 1.7%, constituting a better result than the CT only model that Willems et al. reported. Nevertheless, after comparing the average absolute differences in D_max_ and D_98%_, the results obtained from the PCTGOS model were higher than the CT combined with the contour structure model that Willems et al. reported. Under the circumstances, the 3D U-Net-based deep learning model and 3D input dataset used in their study might perform better when predicting the 3D dose distribution—as opposed to our model, which used 2D images and converted them to 3D images later in the process. Lempart et al. studied the use of the densely connected U-Net deep learning model to predict the dose distribution for VMAT for prostate cancer by using CT images combined with the separated organs’ contours (1PTV + 4OARs + 1Body). After obtaining the predicted dose distribution, they converted it to deliverable treatment plans. They modified the U-Net model to train on triplets data (a combination of three consecutive image slices and corresponding segmentations), resulting in a total of 160 patients whose data were. In this case, they reported that the mean of the absolute differences in D_98%_ of PTV was 1.90%, while for the D_mean_ of the bladder it was 2.1% [23]. Compared to our results, the absolute differences in D_98%_ and D_mean_ from our three models were higher in both parameters. This could be because the input data of our model were less than those used in the model that Lempart et al. reported; the latter model inputted the set of data with a combination of three consecutive slices, which could lead the model to learn the relationships between slices for a higher accuracy in prediction.

Nguyen et al. studied the use of a U-Net-based deep learning model to predict the dose distribution for the IMRT plan for prostate cancer from the contours of organ structure. They reported that the means of the absolute differences in D_max_ and D_mean_ were still less than 5% of the prescription dose in the PTV and OARs [5]. Murakami et al. developed a fully automated dose distribution prediction for the IMRT plan for prostate cancer using the GAN from CT images alone, and reported that the means of the absolute differences in D_max_ and D_mean_ were both less than 2% in PTV [4]. Compared to our results, the differences in D_max_ and D_mean_ from our model were slightly higher than the results of Nguyen et al. and Murakami et al. in both PTV and OARs. This could be because our model predicted the dose distribution from the VMAT plan, not the IMRT plan. Unlike the IMRT, VMAT dose distribution does not have a clear radiation beam path, which could make it harder for the deep learning model to catch the pattern in VMAT dose distribution, leading to worse prediction results and higher uncertainty.

Compared to the traditional process of radiation therapy, our method gives a faster usage time in the treatment planning process, as shown in Figure 6. The traditional treatment planning process may take several days to obtain the desired dose distribution. In contrast, our deep-learning-based dose prediction model could shorten the time from days to 3.5 s for the PCT alone PCTGOS models, and 18 s for the PCTSOS model. In particular, as the PCT alone model does not require physicians to provide the information on contouring structure, this model could shorten the time by even more than other models.

Future work will increase the number of patients’ data used, so as to achieve comparability with previous studies. All three contours of PTV volumes will be implemented in future studies. Additionally, the 3D convolutional neural network model will also be implemented in order to prevent errors from lack of dose continuity between slices. In future investigation, a fully automated treatment planning system will be developed from the accuracy dose prediction based on deep learning for clinical use in medical oncology and radiation treatment.

## 5. Conclusions

The three dose distribution predictive models of the prostate VMAT plan were developed using a generative adversarial network with different input data. Additionally, using our trained models, the accurate and rapid VMAT dose distributions were generated directly from either CT alone or CT and patient organ structure. The mean prediction time was approximately 3.5 s per patient, except with the PCTSOS model, which required approximately 17.5 s per patient. The highest 3D gamma passing rate was 80.51 ± 5.94, and the lowest overall percentage difference in DVH parameters is 6.01 ± 5.44% from the PCTGOS model. From 26 evaluated DVH parameters, the PCTSOS model received the most parameters with the lowest percentage differences (11 parameters; 6 PTV and 5 OAR). This dose prediction model could accelerate the time used for the structural contouring and the iterative optimization process for VMAT treatment planning, by guiding the planner with the desired dose distribution.

## Figures and Tables

**Figure 1 life-11-01305-f001:**
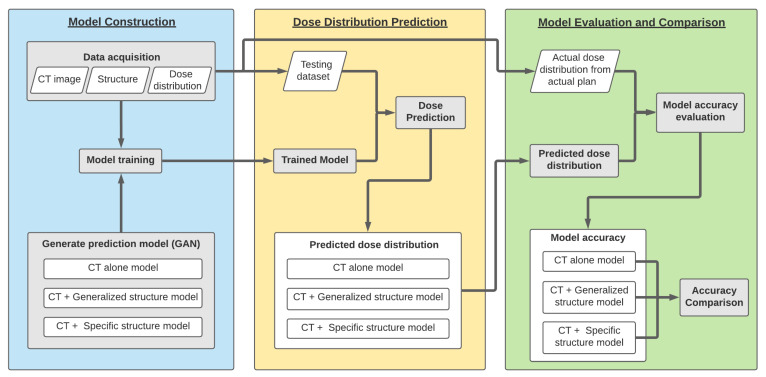
The overall framework of this study. The first process is model construction, the second process is model testing, and the last process is model comparison.

**Figure 2 life-11-01305-f002:**
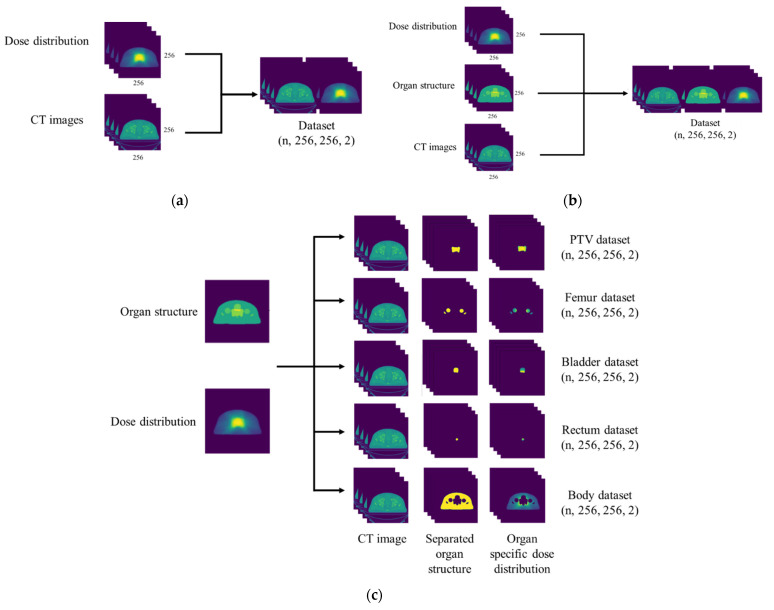
Schematic of data pre-processing and dataset generation: (**a**) dataset for the PCT alone model; (**b**) dataset for the PCTGOS model; (**c**) dataset for the PCTSOS model.

**Figure 3 life-11-01305-f003:**
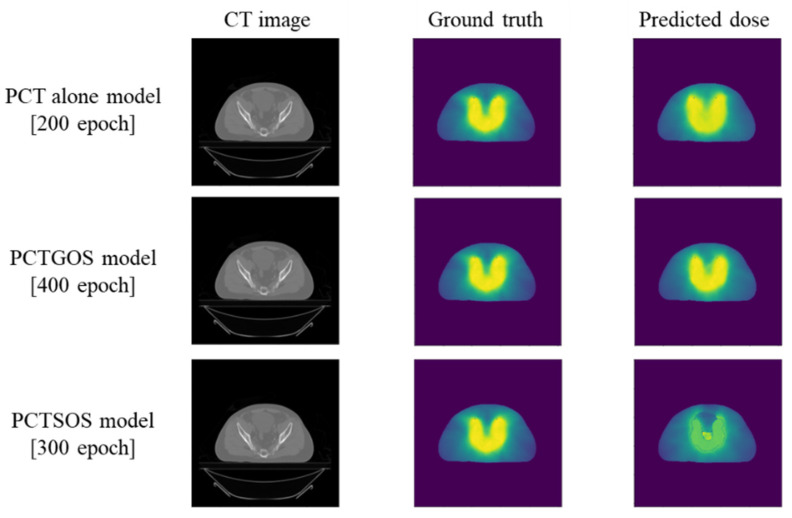
The input CT image, ground truth dose distribution, and predicted dose distribution results of the PCT alone model, PCTGOS model, and PCTSOS model.

**Figure 4 life-11-01305-f004:**
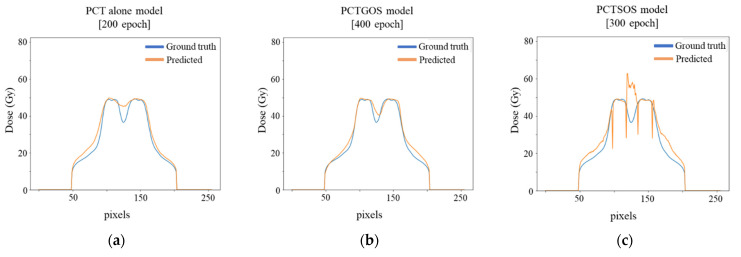
Dose profile comparison between ground truth (blue line) and predicted (orange line) dose distribution for 3 models: (**a**) PCT alone model; (**b**) PCTGOS model; and (**c**) PCTSOS model.

**Figure 5 life-11-01305-f005:**
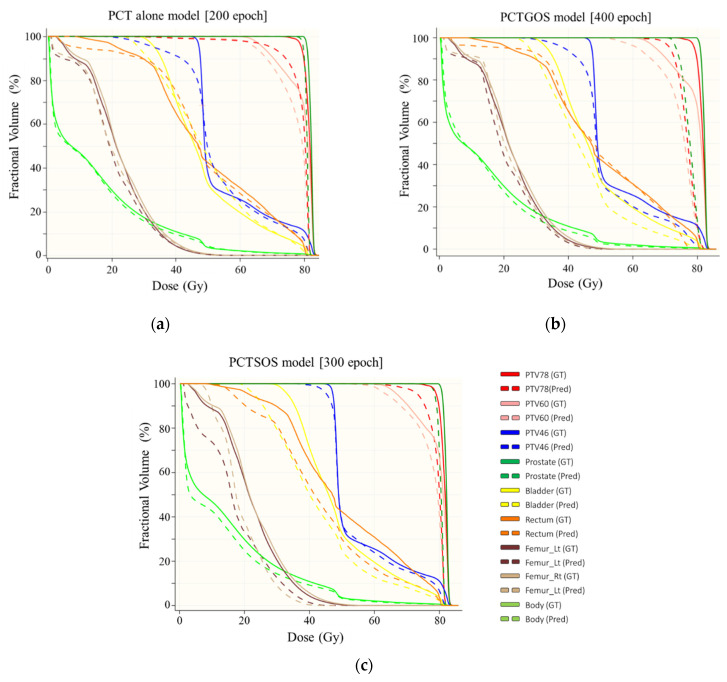
Dose volume histogram results: (**a**) PCT alone model; (**b**) PCTGOS model; and (**c**) PCTSOS model.

**Figure 6 life-11-01305-f006:**
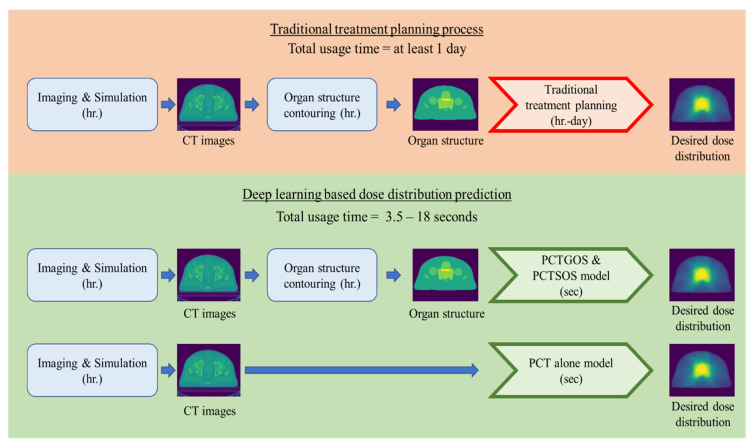
Comparison of a traditional treatment planning process and the deep-learning-based dose distribution prediction.

**Table 1 life-11-01305-t001:** Dose constraints for organs at risk.

Organ	QUANTEC Recommendation	RTOG Recommendation
Bladder	V_60Gy_ < 50%	-
	V_70Gy_ < 35%	-
	V_75Gy_ < 25%	-
Rectum	V_50Gy_ < 50%	-
	V_60Gy_ < 35%	-
	V_65Gy_ < 25%	-
	V_70Gy_ < 20%	-
Femoral heads	-	V_50Gy_ < 5%

**Table 2 life-11-01305-t002:** DVH evaluation parameters for PTV.

Organ	Criteria
PTV78	D_max_
	D_mean_
	D_2%_
	D_95%_
	D_98%_ (near min dose)
PTV60	D_95%_
	D_98%_ (near min dose)
PTV46	D_95%_
	D_98%_ (near min dose)

**Table 3 life-11-01305-t003:** DVH evaluation parameters for OARs.

Organ	Criteria	Recommendation
Bladder	D_max_	
	D_mean_	
	V_60Gy_	QUANTEC
	V_70Gy_	QUANTEC
	V_75Gy_	QUANTEC
Rectum	D_max_	
	D_mean_	
	V_50Gy_	QUANTEC
	V_60Gy_	QUANTEC
	V_65Gy_	QUANTEC
	V_70Gy_	QUANTEC
Left femoral head	D_max_	
	D_mean_	
	V_50Gy_	RTOG
Right femoral head	D_max_	
	D_mean_	
	V_50Gy_	RTOG

**Table 4 life-11-01305-t004:** DVH evaluation parameters for PTV.

Model	Training Time (500 Epoch)	Prediction Time
PCT alone model	~25 h	24.87 s/8 data 3.61 ± 0.19 s/data
PCTGOS model	~25 h	24.60 s/8 data 3.56 ± 0.21 s/data
PCTSOS model	~125 h	123.82 s/8 data 17.48 ± 1.03 s/data

**Table 5 life-11-01305-t005:** Gamma passing rate with 3%/3mm criteria.

Model	Epoch	Average Gamma Passing Rate
PCT alone model	100	72.50 ± 8.95
	200 *	77.21 ± 9.02
	300	76.16 ± 9.01
	400	76.48 ± 8.56
	500	76.09 ± 8.76
PCTGOS model	100	77.29 ± 7.39
	200	79.69 ± 5.80
	300	79.74 ± 5.92
	400 *	80.51 ± 5.94
	500	80.14 ± 5.93
PCTSOS model	100	74.67 ± 3.45
	200	76.45 ± 4.21
	300 *	76.90 ± 3.91
	400	75.12 ± 5.16
	500	74.34 ± 4.65

* Epoch with maximum gamma passing rate for each model.

**Table 6 life-11-01305-t006:** The summary of average percentage differences in DVH parameters between ground truth dose distribution and predicted dose distribution for all three models.

Organ	Criteria	Patient CT Alone Model	Patient CT and Generalized Organ Structure Model	Patient CT and Specific Organ Structure Model
PTV78	D_max_	1.72 ± 1.08	2.95 ± 3.18	1.38 ± 0.83
	D_mean_	5.18 ± 2.26	7.75 ± 7.67	4.82 ± 3.45
	D_2%_	1.00 ± 0.60	2.89 ± 4.04	1.02 ± 0.86
	D_95%_	23.40 ± 13.87	18.82 ± 13.22	16.61 ± 6.32
	D_98%_	32.39 ± 20.25	23.40 ± 16.08	20.87 ± 6.74
PTV60	D_98%_	11.07 ± 10.20	12.90 ± 4.31	10.02 ± 6.69
	D_95%_	16.98 ± 16.84	14.21 ± 3.75	13.05 ± 6.67
PTV46	D_98%_	16.70 ± 20.28	1.92 ± 3.41	7.11 ± 9.20
	D_95%_	19.88 ± 20.27	3.37 ± 5.11	9.39 ± 8.4
Bladder	D_mean_	6.72 ± 6.01	5.59 ± 1.74	4.00 ± 3.20
	D_max_	2.36 ± 3.83	4.26 ± 8.41	0.70 ± 0.46
	V_60Gy_	9.83 ± 7.99	5.52 ± 5.11	4.43 ± 3.15
	V_70Gy_	5.93 ± 5.79	3.83 ± 3.13	2.88 ± 1.85
	V_75Gy_	5.17 ± 4.45	3.87 ± 2.64	2.81 ± 1.57
Rectum	D_mean_	7.98 ± 8.33	4.00 ± 3.59	6.02 ± 3.97
	D_max_	1.19 ± 1.05	4.11 ± 3.25	2.50 ± 1.59
	V_50Gy_	17.09 ± 18.65	9.85 ± 7.84	12.83 ± 5.68
	V_60Gy_	11.67 ± 8.80	4.23 ± 4.00	7.84 ± 4.11
	V_65Gy_	9.25 ± 5.98	3.78 ± 2.99	5.88 ± 3.71
	V_70Gy_	6.88 ± 5.78	3.85 ± 3.00	4.70 ± 2.22
Left femoral head	D_mean_	3.26 ± 2.53	2.64 ± 1.14	3.99 ± 2.14
	D_max_	3.90 ± 2.47	3.59 ± 2.65	7.75 ± 4.15
	V_50Gy_	0.42 ± 0.62	0.48 ± 0.42	0.78 ± 1.16
Right femoral head	D_mean_	2.27 ± 2.28	2.29 ± 0.91	3.62 ± 1.34
	D_max_	8.15 ± 14.44	5.88 ± 2.91	11.28 ± 5.45
	V_50Gy_	0.34 ± 0.39	0.35 ± 0.49	0.63 ± 0.79

**Table 7 life-11-01305-t007:** Summary of the model comparisons.

Comparison Criteria	Patient CT Alone Model	Patient CT and Generalized Organ Structure Model	Patient CT and Specific Organ Structure Model
Prediction time	3.61 ± 0.19 s	3.56 ± 0.21 s	17.48 ± 1.03 s
Max 3D gamma passing rate (3%, 3 mm)	77.21 ± 9.02	80.51 ± 5.94	76.90 ± 3.91
Average %diff over all parameters	8.87 ± 7.74%	6.01 ± 5.44%	6.42 ± 5.08%
Parameters with	5/26	10/26	11/26
minimum average %diff	1 from PTV	2 from PTV	6 from PTV
(From 26 parameters)	4 from OARs	8 from OARS	5 from OARs

**Table 8 life-11-01305-t008:** The prediction performance comparison of our results and the previous studies.

	Model	Data	PTV			Bladder
			D_max_	D_98%_	D_mean_	D_mean_
Our study	GAN	PCT alone	1.72 ± 1.08	16.70 ± 20.28	5.18 ± 2.26	6.72 ± 6.01
		PCTGOS	2.95 ± 3.18	1.92 ± 3.41	7.75 ± 7.67	5.59 ± 1.74
		PCTSOS	1.38 ± 0.83	7.11 ± 9.20	4.82 ± 3.45	4.00 ± 3.20
Willems et al. [22]	3D U-Net (normalized)	CT only	8.6 ± 4.5	16.8 ± 11.5	-	-
		CT + isocenter	6.2 ± 3.4	5.6 ± 3.1	-	-
		CT + contour	1.3 ± 1.3	1.0 ± 2.4	-	-
		CT + isocenter + contour	2.5 ± 1.2	1.6 ± 2.7	-	-
Lempart et al. [23]	Densely connected U-Net	2.5D (3 consecutive slices) CT + 1PTV + 4OARs + 1Body	-	1.90 ± 1.60	-	2.10 ± 3.00
Murakami et al. [4]	GAN	CT-based model (IMRT)	1.68 ± 0.01	-	1.98 ± 0.01	9.14 ± 0.06
Nguyen et al. [5]	U-Net	Structure-based model (IMRT)	1.80 ± 1.09	-	1.03 ± 0.62	4.22 ± 3.63

## Data Availability

Not applicable.

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
