# Peer review of "Predicting Three-Dimensional Dose Distribution of Prostate Volumetric Modulated Arc Therapy Using Deep Learning"

_life, 2021, doi:10.3390/life11121305_

Round 1

Reviewer 1 Report

The ms Predicting three dimensions dose distribution of prostate volumetric modulated arc therapy using deep learning proposes to use Deep Learning AI for planning improvement of 3D dose distribution with VMAT therapy agains prostate cancer.

The manuscript has some challenges in terms of language, particularly of grammar, that need to be addressed and that complicate its readability and understanding. That is in fact, its main downfall. 
Also, to note that the references used are rather scarce when compared with the literature that already exists on the topic, inclusively from the last 2-3 years. 

The work here presented can have significant implications in translational medicine, not only in terms of prostate cancer treatments. However, it requires extensive English language editing to improve readability and clarity. The authors should also make sure that all the acronyms are described. Image legends can be completed with important highlights.
 And at some point, the tissues / organs used to train and the target that the authors want to predict about (prostate) become somehow merged in the text being confusing to the rather to understand exactly what has been done and why. 
Altogether, this work can add to the literature to show how AI can aid in the clinical setting improving treatment options and efficiency in the use of resources. 

pdf in attachment. 

Author Response

Thank you for your constructive comments. I have changed based on your comments both in the comments and in the PDF file. 

Reviewer 2 Report

In this study, Kummanee et al. proposed a deep learning model for Predicting Three Dimensions Dose Distribution of Prostate 2 Volumetric Modulated Arc Therapy. Even though the idea looks ok, some major concerns are raised in this study:

1. The authors didnot have any baseline comparison to show that their model was the optimal one for this specific problem.

2. The authors didnot have any external validation data, which is an important concern for medical studies.

3. How did the authors tune the optimal hyperparameters of the models?

4. Recently, most segmentation models are measured using Dice Loss or IoU. Thus the authors must provide such metrics also.

5. Model is built on 2D or 3D images? The authors should describe it clearly.

6. Deep learning has been used in previous biomedical studies i.e., PMID: 32613242, PMID: 31920706. Thus, the authors are suggested to refer to more works in this description to attract a broader readership.

7. The authors have to compare the predictive performance to previously published works on the same problem/dataset.

8. The authors should report the uncertainties of the models.

9. Source codes should be provided for replicating the study.

Author Response

Thank you for your constructive comments. I have changed based on your comments point by point. Please see attached. 

Reviewer 3 Report

Summary of paper: In this manuscript, the several dose distribution predictive model of VMAT prostate cancer was developed, evaluated, and compared with the core concept of backward planning. This dose prediction model could accelerate the time used for the structure contouring and the iterative optimization process for VMAT treatment planning by guiding the planner with the desired dose distribution.

Major comments:

This article can further discuss the generalization ability of the model.

The reason and innovation of the model used in this manuscript should be further clarified.

The advantages of the model used in this manuscript over others in the field require further explanation.

The author should further clarify why in the PCTSOS model, the dose distribution lacks smoothness due to inconsistent prediction models for specific organs.

Author Response

(The authors gave the same response as above.)

Round 2

Reviewer 2 Report

My previous comments have been addressed well.

Reviewer 3 Report

It can be accepted.